# Machine Learning Project Proposal

## 1   Introduction

This report will describe our proposal for the main project for the course *80245013 - Machine Learning* held at Tsinghua University. The report is divided into Background, Definition, Related Work and the Proposed Method.

## 2   Background

Financial markets are a cornerstone of the global economy, influencing corporate valuations and individual investments alike. With the rise of advanced machine learning algorithms, the challenge of accurately forecasting market behavior has become increasingly relevant. Still, financial data is notoriously difficult to model due to its unpredictable patterns and sudden shifts that traditional methods struggle to capture [6].

In response, Jane Street, a market leader in automated trading[4], has launched a Kaggle competition to develop more accurate and robust prediction models [3]. Solving this challenge could have significant real-world impact by improving trading strategies, leading to more informed decisions and increased profitability. Additionally, the insights gained from applying machine learning to noisy, non-stationary data could enhance the field of time series forecasting, with benefits extending beyond the field of economics and finance to areas such as healthcare, astrology and climate science [5].

## 3   Definition

As our objective is to predict developments in the stock market, there isn't any mathematical definition of our problem. Still, to score highly in the competition we have to create a model which minimizes error. The kaggle competition uses the following error calculation to estimate the scoring in the competition.

$$R^2 = 1 - \frac{\sum w_i(y_i - \hat{y}_i)^2}{\sum w_i y_i^2}$$

Figure 1: The formula used to decide the model placements in the compeition

Here is an explanation of the symbols used:

- $w_i$: A weight applied to each observation $i$ in the dataset, allowing certain observations to have more or less influence on the error calculation, possibly based on relevance or importance.
- $y_i$: The actual observed value for observation $i$ in the dataset, representing real stock market values that the model aims to predict.

Submitted to 38th Conference on Neural Information Processing Systems (NeurIPS 2024). Do not distribute.

- $\hat{y}_i$: The predicted value generated by the model for observation $i$.

This error calculation evaluates the weighted squared differences between actual and predicted values, adjusting the impact of errors by $w_i$ to reflect competition scoring criteria. As of October 30, 2024, the 10 best scores in the competition are in the range `0.0068`-`0.0050`.

## 4 Related Work

**Hybrid Bidirectional LSTMs (H.BLSTMs)**
- **Advantages:** Captures long-term dependencies, adapt to any changing market conditions.
- **Disadvantages:** Complexity and time computation, dependence on data quantity and quality.

**Extended Kalman filter Non-linear Autoregressive Neural Network (EKF-NAR)**
- **Advantages:** Computes with a big improved accuracy and can also handle complex patterns.
- **Disadvantages:** Can barely handle very complex models with some linearization error.

## 5 Proposed Method

Our project will implement Jamba, a novel Mamba-Transformer hybrid machine learning model. We hope that using Jamba we can leverage the strengths of transformer architectures and Mamba, which is optimized for high-dimensional and sequential data. Our choice of Jamba is driven by its ability to capture complex temporal dependencies and its adaptability to high-volume, real-time data, making it a suitable candidate for predicting stock market behavior in this competition. Furthermore it is a newly developed model, only being released this year, in 2024.

### 5.1 Dataset Selection

We will use the provided Kaggle dataset, containing real-world data derived from Jane Streets production systems, as the models primary data source. We anticipate that this dataset will provide sufficient information for training our model, and therefore don't believe we will need to explore supplementary datasets during the project.

### 5.2 Baseline Approaches

In the Kaggle competition, our model's performance will be evaluated against other participants using the scoring formula outlined in section 3. This will enable us to gauge our model's effectiveness relative to the alternative solutions employed by other teams. Consequently, our focus will be on optimizing our model to achieve the highest possible score, rather than conducting comparative analyses against other models independently.

### 5.3 Implementation

Our implementation of the Jamba model on the Kaggle dataset will follow these steps:

1. **Data Preprocessing:** Raw data will be cleaned, scaled, and organized to remove noise and manage missing values. For time-series data, we will create feature windows capturing recent past values as input to the model.
2. **Model Training:** Jamba will be trained on this data, with the training process involving cross-validation to optimize hyperparameters such as learning rate, sequence length, and transformer depth.
3. **Evaluation Metrics:** We will use the competition's scoring metric, $R^2$, as the primary metric, but may also use MAE (Mean Absolute Error) and MSE (Mean Squared Error) for additional insights.

Our approach may evolve based on preliminary results, but this outline provides a structured plan for the method.

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
