# OpenReview forum: "[Proposal-ML] *****"
_tsinghua.edu.cn/THU/2024/Fall/AML — THU 2024 Fall AML Submission_

### Official Review · ~Peidong_Zhang1 · 2024-11-08
**Strengths and limitations of proposal**

**Rating:** 8
**Confidence:** 3

**Review:**

This proposal outlines the development of the Jamba model, a novel hybrid Mamba-Transformer architecture, for stock market prediction in the Kaggle competition organized by Jane Street. The project aims to leverage Jamba’s ability to capture complex temporal dependencies and handle high-dimensional, real-time data, making it suitable for financial forecasting. However, the proposal does not adequately address potential overfitting issues or the computational complexity of combining these two architectures. The reliance solely on the Kaggle dataset without considering external datasets might also limit the model's robustness and generalizability across diverse market conditions. Finally, the absence of a detailed risk analysis or mitigation strategies for data noise and missing values could hinder the model's ability to handle real-world financial unpredictability effectively.

---

### Official Review · ~Yanchen_Wu1 · 2024-11-08
**Classical Problem**

**Rating:** 7
**Confidence:** 4

**Review:**

This work hopes to achieve better predictions by improving the structure of Mamba-Transformers. The problem studied is the financial data on kaggle, which is a very classic problem. I hope the authors can develop some new methods to achieve better results.

---

### Official Review · ~Xiying_Huang2 · 2024-11-09
**Review of “Machine Learning Project Proposal”**

**Rating:** 7
**Confidence:** 4

**Review:**

This proposal outlines a promising and innovative approach to stock market forecasting by implementing the Jamba model, which integrates transformer and Mamba architectures. The methodology is well-structured and geared toward optimizing performance in a Kaggle competition, with clear steps for data preprocessing, model training, and evaluation. The proposal’s relevance to financial forecasting highlights its potential real-world impact, especially in handling complex, high-dimensional data. However, further clarification on the unique strengths of the Jamba model and a more detailed plan for managing financial data challenges would strengthen its contribution.

---

### Official Review · ~Yufei_Zhuang1 · 2024-11-09
**Shows great promise**

**Rating:** 7
**Confidence:** 4

**Review:**

The choice to rely solely on the provided Kaggle dataset from Jane Streets' production systems is reasonable given their belief in its sufficiency. This simplifies the data - related complexity and allows the team to focus more on model development.
The evaluation approach of using the competition's scoring formula to measure against other participants is practical. It streamlines the process of determining the model's effectiveness without getting bogged down in extensive comparative analyses of other models independently.
The implementation plan is structured and comprehensive. The data preprocessing steps, including cleaning, scaling, and handling time - series data with feature windows, are standard yet crucial for good model performance. The training process with cross - validation for hyperparameter optimization shows a solid understanding of model tuning. Using the competition's scoring metric R2 as the primary evaluation metric, along with potential use of MAE and MSE, provides a good balance in assessing the model's quality. Overall, the proposed method appears to be a strong contender in the Kaggle competition.

---

### Official Review · ~Yuanda_Zhang1 · 2024-11-09
**Kaggle competition**

**Rating:** 7
**Confidence:** 4

**Review:**

The proposal outlines a project aimed at predicting stock market behavior using a novel Mamba-Transformer hybrid model named Jamba. The project is motivated by the challenges of accurately forecasting financial markets and aims to participate in a Kaggle competition organized by Jane Street to develop robust prediction models.
The proposal addresses a significant and relevant challenge in financial forecasting, which has real-world implications for trading strategies and investment decisions. However, it would benefit from a deeper exploration of the model's adaptability to financial data challenges and a more comprehensive validation strategy beyond competition performance.

---

### Official Review · ~Kehan_Zheng1 · 2024-11-11
**Nice Proposal**

**Rating:** 7
**Confidence:** 4

**Review:**

The proposal presents an interesting approach to stock market prediction using a hybrid model called Jamba, which combines Mamba and Transformer architectures. The structure of the proposal is clear, covering background, problem definition, methodology, and evaluation metrics, aiming to solve traditional machine learning problems. However, the project could be further enriched by exploring more creative solutions or by applying the model to datasets or scenarios with greater real-world relevance. This would not only add depth to the analysis but also enhance the project’s overall impact and meaningfulness.

---

### Official Review · ~Wenjing_Wu1 · 2024-11-11

**Rating:** 7
**Confidence:** 3

**Review:**

**Summary**:
The proposal outlines a Mamba-Transformer hybrid model designed to handle high-dimensional and sequential data

**Strengths**:
- Innovative Model Design: Combining Mamba and Transformer architectures could enhance temporal pattern recognition and adaptability to real-time data.
- Clear, Structured Methodology: The plan’s structure provides a comprehensive roadmap, from data preparation to evaluation metrics.

**Weaknesses**:
- Competition-based: Relying solely on the provided dataset may not be able to verify the capability of newly-proposed method, which would restrict the impact of it.

---

### Official Review · ~Yida_Lu1 · 2024-11-11
**A practical method of a competition**

**Rating:** 7
**Confidence:** 4

**Review:**

This study attempts to use Jamba, a newly proposed machine learning structure which is a hybrid of Transformer and Mamba, to make predictions on financial datas in a Kaggle competition. The task is well-defined and the method is clear. On the other hand, this study may be further strengthened by comparing with the traditional Transformer models to demonstrate the effectiveness of Jamba. And there is also space to explore some innovative methods beyond simply fine-tuning to achieve a better performance.

---

### Official Review · ~Ruowen_Zhao1 · 2024-11-11
**Summary and Concern**

**Rating:** 7
**Confidence:** 4

**Review:**

### Summary
The authors describe a structured approach to implement the Jamba model on the Kaggle dataset. Their process involves three main stages: data preprocessing, model training, and evaluation.

### Concern
+ Lack of data analysis: The outlined approach lacks a thorough analysis of the input data. Without this step, the preprocessing and model training stages may miss important characteristics of the data.
+ Lack of analysis on training model selection: The approach does not provide a detailed analysis for choosing the Jamba model as the primary training model. Without evaluating alternative models or explaining why Jamba is the most suited to the dataset’s characteristics, there’s a risk that the model might not be the most effective for this task.

---

### Official Review · ~Ziyu_Zhao6 · 2024-11-11

**Rating:** 7
**Confidence:** 3

**Review:**

Overview:
This proposal presents a project aimed at improving financial market forecasting through a novel hybrid model called Jamba—a Mamba-Transformer model optimized for handling high-dimensional, sequential data in real-time. Leveraging recent advancements in machine learning, particularly in time-series prediction, this project participates in Jane Street’s Kaggle competition to enhance trading strategy models.

Strengths:
	1. Innovative Model Choice: The use of a hybrid Mamba-Transformer model is an intriguing choice, as it combines the temporal dependency-capturing strengths of Transformers with the sequential data capabilities of Mamba. This is well-suited for the dynamic, high-dimensional data seen in financial markets.
	2. Clear Methodology: The structured steps—from data preprocessing to evaluation—offer a clear roadmap for implementation, which can help in achieving reproducible results. The use of cross-validation and established error metrics like R2, MAE, and MSE is a solid approach to ensure reliability.

Weaknesses:
	1. Potential Data Limitations: By relying solely on the provided Kaggle dataset, the project might miss out on additional data sources that could enrich training and generalization. Exploring supplementary datasets could further improve the model’s adaptability to diverse financial conditions.
	2. Risk of Overfitting: Given the competitive context and focus on optimizing R2 for the Kaggle score, there is a risk of overfitting to the dataset and scoring metric. This focus could limit the model’s generalizability to unseen financial data, so regularization strategies or further evaluation metrics might be beneficial.

---

### Official Review · ~Bowen_Gao1 · 2024-11-12

**Rating:** 7
**Confidence:** 4

**Review:**

**Summary**

This proposal addresses the Jane Street stock market prediction challenge on Kaggle, where the authors suggest using "Jamba," a novel Mamba-Transformer hybrid machine learning model, to improve prediction performance.

**Strengths:**

1. The motivation for using Jamba is well-justified, as its ability to capture complex temporal dependencies and adaptability to high-volume, real-time data aligns with the demands of stock market prediction.
2. The proposal provides a clear outline of the pipeline, covering key elements such as model training, dataset selection, and model evaluation.

**Weaknesses:**

1. The proposal lacks a correct title that is relevant to the content of the proposal.
2. The proposed method is relatively simplistic, involving a straightforward application of an existing model to the task without significant adaptation or innovation.

---

### Official Review · ~jin_wang30 · 2024-11-12

**Rating:** 7
**Confidence:** 4

**Review:**

Overall, the article presents an innovative machine learning project proposal in the field of financial forecasting. The article explains the financial market background, data selection, model architecture innovation and target evaluation indicators at a high level, which illustrates the theoretical value and practical application potential of the model.


The article clearly organizes the content into sections such as "Background", "Definition", "Related Research" and "Proposed Method", making it easy for readers to understand how each link connects and supports the project goals. This structure is particularly suitable for complex machine learning project proposals because it not only presents the background and motivation, but also briefly introduces the implementation process of the project. In addition, the article details the challenges of financial market forecasting in the "Background" section, including data noise, instability and other issues, and mentions the actual impact of forecasting. This provides a reasonable background for the subsequent proposed methods and highlights the practical application value of this project, such as improving trading strategies and providing investment decision support. In addition, the article extends the prediction problem of financial markets to a wider range of time series prediction problems, such as healthcare and climate science, emphasizing the universality of the method. The article details the scoring criteria (R2) of the competition and states that this will be used as the main evaluation metric of the model, while MAE and MSE may be used as auxiliary indicators. This variety of evaluation index design helps to measure model performance from different angles and ensure that the model meets high standards in terms of accuracy and error control.


However, the article also has shortcomings. The first shortcoming is that there are many format problems in the article, which does not meet the requirements of this assignment. The second shortcoming is that the article is too weak in innovation, and only uses the large model Jamba without proposing any ideas for improvement.